# The TLK1–MK5 Axis Regulates Motility, Invasion, and Metastasis of Prostate Cancer Cells

**DOI:** 10.3390/cancers14235728

**Published:** 2022-11-22

**Authors:** Md Imtiaz Khalil, Arrigo De Benedetti

**Affiliations:** Department of Biochemistry and Molecular Biology, Louisiana State University Health Shreveport, Shreveport, LA 71103, USA

**Keywords:** TLK1/1B, MAPKAPK5/MK5/PRAK, FAK, paxillin, HSP27, ERK3, prostate cancer metastasis

## Abstract

**Simple Summary:**

Recent work by us and others have illustrated the critical importance of MK5/PRAK in the invasive and motility properties of several cancer cell lines and some mouse models. In our earlier work, we also uncovered that TLK1 modulates the activity of MK5 by phosphorylating S354 and two additional sites (S160 and S386). We have now expanded on the possible mechanisms of the TLK1 > MK5 axis mediated pro-motility and invasive activity and report that this may be due to reorganization of the actin fibers at lamellipodia and the focal adhesions network, in conjunction with increased expression of some MMPs. Pharmacological or genetic manipulation of prostate cancer (PCa) cell lines, LNCaP and PC3, results in drastic loss of in vitro motility and invasive capacity of these cells concomitant with alterations of their general morphology and reorganization of the focal adhesions distribution. In addition, PC3 used in tail-vein experimental metastases studies shows that the use of GLPG0259 (MK5 inhibitor) or J54 (TLK1 inhibitor) results in a drastic reduction of metastatic lung nodules, both macroscopically and histologically.

**Abstract:**

**Background**: Metastatic dissemination of prostate cancer (PCa) accounts for the majority of PCa-related deaths. However, the exact mechanism of PCa cell spread is still unknown. We uncovered a novel interaction between two unrelated promotility factors, tousled-like kinase 1 (TLK1) and MAPK-activated protein kinase 5 (MK5), that initiates a signaling cascade promoting metastasis. In PCa, TLK1–MK5 signaling might be crucial, as androgen deprivation therapy (ADT) leads to increased expression of both TLK1 and MK5 in metastatic patients, but in this work, we directly investigated the motility, invasive, and metastatic capacity of PCa cells following impairment of the TLK1 > MK5 axis. **Results**: We conducted scratch wound repair and transwell invasion assays with LNCaP and PC3 cells to determine if TLK1 and MK5 can regulate motility and invasion. Both genetic depletion and pharmacologic inhibition of TLK1 and MK5 resulted in reduced migration and invasion through a Matrigel plug. We further elucidated the potential mechanisms underlying these effects and found that this is likely due to the reorganization of the actin fibers at lamellipodia and the focal adhesions network, in conjunction with increased expression of some MMPs that can affect penetration through the ECM. PC3, a highly metastatic cell line when assayed in xenografts, was further tested in a tail-vein injection/lung metastasis model, and we showed that, following inoculation, treatment with GLPG0259 (MK5 specific inhibitor) or J54 (TLK1 inhibitor) resulted in the lung tumor nodules being greatly diminished in number, and for J54, also in size. **Conclusion**: Our data support that the TLK1–MK5 axis is functionally involved in driving PCa cell metastasis and clinical aggressiveness; hence, disruption of this axis may inhibit the metastatic capacity of PCa.

## 1. Introduction

Prostate cancer (PCa) is one of the most frequently diagnosed diseases in men in the Western world. Although it is a slow-growing malignancy, the mortality rate from this cancer is ranked second among all the male-specific diseases in United States [1]. The advent of first- and second-generation anti-androgens have reduced PCa-specific deaths by a great extent; however, recent statistics suggest that the death rate from PCa is gradually increasing [1,2,3,4]. Therapy resistance and, subsequently, PCa aggressiveness (hormone refractoriness) and metastasis could account for this gradual increase in PCa-related deaths. Hence, there is a crying need to identify newer clinically relevant targets and devise medications which can be combined with present therapeutic modalities for the treatment of metastatic castration-resistant prostate cancer (mCRPC).

Protein kinases participate in numerous cellular processes, including cell motility, migration, and invasion, through the reversible phosphorylation of their direct and indirect substrates. The deregulation of protein-kinase-mediated signal transductions is evident in various types of cancers, and, thus, these could be effective targets for the clinical management of carcinogenesis and metastasis. Although there are some caveats, protein kinases can be selectively and effectively targeted by specific small molecule inhibitors, which bind to their ATP binding sites or, more preferentially, an allosteric site, leading to the functional inhibition of the enzymes (reviewed in References [5,6]).

We previously demonstrated that a serine/threonine (S/T) protein kinase, Tousled-like kinase 1 (TLK1), could promote PCa cell motility and clinical aggressiveness by phosphorylating another S/T kinase, MAPK-activated protein kinase 5 (MAPKAPK5/MK5) [7,8]. TLK1 is known as a genome maintenance regulator which functions in regulating replication, transcription, cell-cycle checkpoint control, and DNA-damage response and repair through its direct and indirect substrates (reviewed in References [9,10,11,12]). TLK1′s role in driving PCa progression is an active pursuit of our lab. We have demonstrated that TLK1 can promote androgen-dependent to androgen-independent progression and survival of PCa cells through various mechanisms, such as hyperactivating NEK1 and NEK1-mediated activation of ATR > Chk1 or VDAC1, stabilization of YAP1, or by directly regulating AKT through AKTIP phosphorylation [13,14,15,16,17,18]. TLK1-driven motility promotion of PCa cells was a novel finding from our lab, although it was known that TLK2, a closely related kinase of TLK1 (both shares 96% homology in their kinase domain and 84% overall homology), can increase the migration and invasion of breast cancer and glioblastoma cell lines [19,20]. On the other hand, MK5 is a pro-metastatic factor that drives the motility of both non-malignant and neoplastic cells through actin cytoskeletal reorganization and focal adhesion proteins modulation [21,22,23,24,25]; however, a couple of studies also documented its anti-motility function [26,27]. MK5′s function may thus depend on the cellular context, type, stage, and/or specific stimulation in cancer. This is particularly important in PCa, as androgen-deprivation therapy (ADT) translationally upregulates TLK1B (a spliced variant of TLK1 with intact kinase domain as TLK1) through the activation of the AKT > mTORC1 > 4EBP1 > eIF4E pathway (reviewed in Reference [9]). In our previous study, we demonstrated that TLK1/1B can directly phosphorylate MK5 in three novel residues (S160, S354, and S386) and increase MK5′s catalytic activity toward its downstream substrates [7]. In fact, TLK1 alone cannot promote cellular motility in the absence of MK5, as observed in a TLK1-overexpressing MK5^−/−^ MEF cell line model. We showed that a non-phosphorylatable S354A mutation of MK5 impairs its motility function like a kinase-dead MK5 (K51E) in MEF cells. We found that ADT increases the pMK5 S354 level and TLK1 inhibition reduces it, thus establishing TLK1 as an authentic MK5 S354 kinase. Moreover, pMK5 S354 was elevated in the tissue sections of high-grade clinical samples and in a transgenic mice model of PCa (TRAMP), which also suggested that the pMK5 S354 level might be associated with the increased aggressiveness of PCa. Finally, we observed that pharmacologic inhibition of MK5 leads to a drastic reduction of motility in a panel of androgen-dependent or -independent PCa cell lines, even at a low dose (1–5 µM concentration) [7]. 

In this present study, we aimed to investigate the mechanisms of TLK1–MK5 signaling in PCa cell migration and invasion. We confirmed the involvement of TLK1 and MK5 activity in enhancing motility and invasion by genetically depleting both kinases in a metastatic PCa cell line. We demonstrated that pharmacologic inhibition/genetic depletion of the effector kinase: MK5 triggers the reorganization of actin filaments from the migratory edge to basal surface of the cells, most probably, by post-translational modulation of focal contact proteins (FAK, paxillin) and HSP27. In addition, MK5 inhibition leads to ERK3 destabilization, which downregulates several matrix metalloproteinases (MMP2 and MMP9) that impair the invasion capacity of the cells. Finally, pharmacologic inhibition of TLK1 or MK5 was demonstrated to be equally effective for the prevention of PCa cell metastasis in vivo. 

## 2. Materials and Methods

### 2.1. Plasmids and Antibodies

Scrambled, TLK1-, and MK5-specific shRNA constructs in lentiviral GFP vector were purchased from Origene (Rockville, MD, USA, TLK1 shRNA cat# TL320623, MK5 shRNA cat# TL320583, scrambled shRNA cat# TR30021). The following primary antibodies were used in this study: rabbit anti-TLK1 (ThermoFisher, Waltham, MA, USA, cat# 720397), mouse anti-PRAK/MK5 (Santa Cruz Biotechnology, Dallas, TX, USA, cat# sc-46667), rabbit anti Phospho- FAK Y397 (ThermoFisher, cat# 44-624G), rabbit anti-Phospho- FAK Y861 (abcam, Cambridge, MA, USA, cat# ab4804), rabbit anti-FAK (Cell signaling technology, Danvers, MA, USA, cat# 3285S), rabbit anti-phospho-paxillin Y118 (Cell signaling technology, cat# 2541S), rabbit anti-paxillin (Santa Cruz Biotechnology, cat# sc-5574), rabbit anti-phospho-HSP27 S82 (ThermoFisher, cat# 44-534G), mouse anti-HSP27 (ThermoFisher, cat# MA3-015), rabbit anti- ERK3 (Cell signaling technology, cat# 4067S), mouse anti-MMP2 (Santa Cruz Biotechnology, cat# sc-13595), mouse anti-MMP9 (Santa Cruz Biotechnology, cat# sc-13520), mouse anti-MMP3/10 (Santa Cruz Biotechnology, cat# sc-374029), mouse anti-Ki-67 (Cell signaling technology, cat# 9449S), and rabbit anti-GAPDH (Cell signaling technology, cat# 2118S). 

### 2.2. Cell Culture

LNCaP and PC3 cells were cultured in RPMI 1640 media containing 10% FBS and 1% antibiotic–antimycotic solution. All the cells were maintained in a humidified incubator at 37 °C with 5% CO_2_. These cells were purchased from American Type Culture Collection (ATCC, Manassas, VA, USA) and were authenticated within the past three years.

### 2.3. Lentiviral Transduction

Lentiviral packaging of scrambled (shSCR), TLK1- or MK5-specific shRNA plasmids were performed in the LSU Health Shreveport COBRE core facility. PC3 parental cells were infected either with scrambled, TLK1, or MK5 shRNA containing lentivirus in a 6-well plate when the cells reached 50–60% confluency. First, culture media was removed, and 1 mL of viral supernatant was added to each well with 10 ug/mL of polybrene (Millipore sigma, Saint Louis, MO, USA, cat# TR-1003-G) to enhance the infection efficiency. After 3 h, 1 mL of fresh culture media was added to each well, and the cells were incubated with viral supernatant containing media for 48–72 h. During this time, cells were monitored under a fluorescent microscope for the GFP signal to ensure successful infection. After the incubation period, viral supernatant containing media was removed, and cells were selected with 1 ug/mL of puromycin (Sigma-Aldrich, Saint Louis, MO, USA, cat# P8833) treatment for at least seven days. 

### 2.4. Cell Treatment

LNCaP and PC3 cells were treated with 1–5 µM of GLPG0259 (hereafter GLPG; Medkoo Biosciences, Inc., Morrisville, NC, USA, cat# 561481) for 48 h in a T-25 flask when cell confluency reached 70–90%. DMSO-treated cells were considered to be the vehicle control (VC). After the treatment, cells were harvested for Western blotting (WB) and qPCR analysis. 

### 2.5. Scratch Wound Repair Assay

Scratch wound repair assays, using either scrambled shRNA containing, TLK1, or MK5 knocked-down PC3 cells, were conducted as previously described [7].

### 2.6. Trans-Well Invasion Assay 

Trans-well invasion assay was conducted by using Incucyte^®^ S3 Live-Cell Analysis Instrument (Sartorius, Bohemia, NY, USA), following the manufacturer’s protocol. Briefly, both sides of the porous membrane of the top insert of Incucyte Clearview 96-well chemotaxis plate (Sartorius, Bohemia, NY, USA, cat# 4582) were coated with 50 µg/mL of Matrigel (Corning, Glendale, AZ, USA, cat# 356234) and incubated for 1 h at 37 °C. Scrambled shRNA, TLK1, or MK5 knocked-down PC3 cells were seeded in the top chamber as 1000 cells/well in no serum media. To determine the effect of pharmacologic inhibition of MK5 in cell invasion, PC3 parental cells were seeded with no-FBS media containing either DMSO as the vehicle control (VC) or 0.1–5 µm of GLPG. Medium containing 20% FBS + 100 ng/mL of EGF was provided in the bottom reservoir wells to create the chemotactic gradient. Cells were then transferred to an Incucyte S3 incubator with 5% CO_2_ at 37 °C, and the invasion of the cells toward chemotactic gradient was monitored by the Incucyte S3 live-cell analysis system. Images were taken at every 4 h of both the top insert and bottom chamber, and the total cell occupancy in the bottom chamber normalized to the initial top value was determined by the Incucyte software (version 2020A). Mean rate (MR) was determined by calculating the slopes of the curve of each well and used to conduct statistical analysis. 

### 2.7. Immunocytochemistry (ICC) and Actin Staining

Parental or MK5 knocked-down PC3 cells were seeded in a Falcon^®^ 4-well Culture Slide (Corning, cat# 354104) and grown till 50–60% confluency before starting the treatment and ICC procedure. Parental PC3 cells were treated with 1–5 µM of GLPG (Medkoo Biosciences, cat# 561481) for 48 h. For the detection of endogenous paxillin and F-actin, cells were fixed in 2% formaldehyde solution by adding equal amount of 4% formaldehyde (Santa Cruz Biotechnology, cat# sc-281692) directly in the culture media and incubated for 20 min at room temperature. Next, the formaldehyde was aspirated out, and the cells were then permeabilized in 0.5% Triton X-100 (Fisher Scientific, Pittsburgh, PA, USA, cat# BP151–100) for 10 min. Cells were washed thrice with TBST and blocked in 1% BSA with sodium azide solution for 1 h at room temperature. Cells were washed three times and incubated with diluted rabbit anti-paxillin (Santa Cruz Biotechnology, cat# sc-5574) primary antibody overnight at 4 °C. After rinsing and washing off the primary antibody, cells were incubated with Alexa Fluor 488 goat anti-rabbit IgG antibody (ThermoFisher, cat# A-11034) and diluted rhodamine-labeled phalloidin (ThermoFisher, cat# R415) containing 1% BSA for 1 h at room temperature. For F-actin staining alone, GLPG treated parental and MK5 depleted PC3 cells were fixed in 4% formaldehyde solution for 15 min at room temperature, followed by the permeabilization in 0.1% Triton X-100 for 15 min and incubation in rhodamine phalloidin with 1% BSA at room temperature for 1 h. Cells were then counterstained and coverslipped, using Vectashield antifade mounting media with DAPI (Vector laboratories, Burlingame, CA, USA, cat# H-1200). F-actin and paxillin localization were imaged by using a Nikon A1R Confocal microscope.

### 2.8. Western Blotting

Cells were harvested, and the lysate was prepared in 1X RIPA lysis buffer (Santa Cruz Biotechnology, cat# sc-24948) by sonication, followed by the measurement of protein concentration, using BCA Protein Assay Kit (ThermoFisher, cat# 23225). Cell lysates were run in either 7.5% or 12% SDS–PAGE gel and then transferred into a PVDF Immobilon-P 0.45 µm membrane (Millipore Sigma, Saint Louis, MO, USA, cat# IPVH00010). Membranes were blocked in 5% nonfat dry milk and then incubated in primary antibodies overnight at 4 °C. After washing, membranes were incubated in horseradish peroxidase conjugated secondary antibodies for 1 h at room temperature and finally developed by using Bio-Rad ChemiDoc Imaging System (Bio-Rad, Hercules, CA, USA, cat# 12003154), using ECL substrates (ThermoFisher, cat# 32106). Densitometric quantifications of band intensity were conducted by using ImageJ (version 1.48v) software. 

### 2.9. Real-Time Quantitative PCR (RT-qPCR)

Total RNA was extracted by using TRIzol reagent (ThermoFisher, cat# 15596026), following manufacturer’s protocol. The RNA concentration was determined by using Nanodrop, and 1 µg of RNA/reaction was used to synthesize Complementary DNA (cDNA), using ProtoScript First Strand RNA synthesis reverse transcriptase and oligo (dT) primers (New England Biolab, Ipswich, MA, USA, cat# E6300L). Then qPCR was conducted, using iQ SYBR green supermix (Biorad, cat# 1708880, Des Plaines, IL, USA) and Bio-Rad CFX96 Fast Real-Time PCR Systems. Gene-expression changes were determined by using the ∆∆Ct relative quantification method. GAPDH mRNA was used as an internal control. All values are presented as mean ± standard error of mean (SEM). Table 1 contains the primers list and the sequences used in this study.

### 2.10. Tail-Vein Injection

Fox Chase SCID Beige male mice (CB17.Cg-*Prkdc^scid^Lyst^bg-J^*/Crl) was purchased from Charles River laboratory (Wilmington, MA, USA). All mice were kept in the pathogen-free animal facility of LSU Health Shreveport. All experimental procedures were approved by LSU Health Shreveport Institutional Animal Care and Use Committee (IACUC). 1 × 10^6^ PC3 parental cells were resuspended in 100 µL of PBS and injected into the lateral tail vein of each mouse at 5 weeks of age. Within a day of tail-vein injection, intraperitoneal administration of GLPG and J54 was started twice a week for 7 weeks. GLPG and J54 were administered at a dose of 2 mg/kg and 10 mg/kg of mouse body weight, respectively. Mice were monitored for clinical signs of metastasis, and their body weight was measured every two days. After seven weeks, mice were euthanized, and the lungs were collected and fixed in formalin. A 0.5× Olympus microscope was used for capturing representative lung images from each group, and tumor nodules were counted. 

### 2.11. Immunohistochemistry (IHC)

IHC staining of mice lung tissue was conducted as previously described [7]. Imaging was performed by using an Olympus BX43 microscope at 4× and 10× magnification. 

### 2.12. Statistical Analysis

GraphPad Prism 9 (GraphPad Software Inc., San Diego, CA, USA) and Microsoft excel software (version 2210) (Microsoft Corporation, Redmond, WA, USA) were used for statistical analysis, and GraphPad Prism 9 was used for plotting the graphs. Data quantifications are expressed as mean ± standard error of the mean (SEM). For the comparison of multiple groups, one-way ANOVA, followed by Tukey’s post hoc analysis, was conducted; *p*-values < 0.05 were considered significant.

## 3. Results

### 3.1. Knockdown of TLK1 or MK5 Results in Reduced Motility and Invasive Capacity of PC3 Cells

We previously reported that the TLK1 > MK5 interaction is critical for the motility and invasive properties of all the PCa cell lines we tested, and this was most critically illustrated with the use of the specific MK5 inhibitor GLPG (in a dose-dependent manner) [7]. However, as in all instances of pharmacological inhibition, there was an issue of ascertaining specificity—both for MK5 and for its regulation/activation by TLK1. To circumvent this potential problem, we first tested how stable knockdown of TLK1 with shRNA can affect the motility of PC3 cells—note that we were unsuccessful in obtaining a similarly knocked down line of LNCaP cells, for which apparently TLK1 is essential. Four distinct shRNAs in lentivirally infected PC3 cells clonal pools were examined, and two demonstrated a strongly reduced expression of TLK1 and TLK1B splice variants, confirming the effectiveness of the shRNA (Figure 1A), and their expression level was quantified (Figure 1B). To determine the resulting effect on motility and invasion, we used two features of the Incucyte system: scratch assay; and invasion through an insert coated with Matrigel, as visualized through two focal planes, respectively (Figure 1C–F). In both assays, the two clones with significantly lower TLK1/1B expression showed much reduced motility and invasion (very significant especially for invasion). 

In order to establish the direct contribution of MK5 in the presence of TLK1 component and to confirm the effect of GLPG-mediated inhibition of MK5 on motility, we performed the same two assays in PC3 cells that were stably knocked down for MK5 (Figure 1G,H). Four different shRNAs to MK5 were used to infect PC3 cells, and three resulted in a highly significant reduction of MK5 expression. All three stably knocked-down clones showed significantly reduced properties of motility and invasion through Matrigel plugs (Figure 1I–L). To determine if PC3 invasive properties could be directly affected via pharmacologic inhibition of MK5, we also tested GLPG on the invasion assay through a Matrigel plug with the Incucyte. Indeed, GLPG abolished the invasion of PC3 cells in a dose-dependent fashion (Appendix A). 

### 3.2. Treatment of LNCaP and PC3 Cells with GLPG Results in Actin Fibers’ Relocalization, Which Likely Affects Motility

To test one of our key hypotheses that the TLK1 > MK5 axis promotes motility through enhancing the formation of lamellipodia via the increased formation of the actin stress fibers’ network, and that inhibiting this axis will impair motility/metastasis, we treated LNCaP and PC3 cells with GLPG. The staining of F-actin with rhodamine-conjugated phalloidin of VC-treated cells showed an intense actin network at the periphery and lamellipodial region of spindle-shaped cells, with some concentrated foci at focal adhesions, in both LNCaP and PC3 cells. The treatment of LNCaP cells with GLPG resulted in a progressive loss of the actin fibers from the peripheral lamellipodia and abrogation of spindliness of the cells to a more or less spherical shape, in a dose-dependent fashion (Figure 2A). Similarly, PC3 GLPG-treated cells showed a dose-dependent loss of actin fibers along the periphery, loss of the spindle morphology concomitant with enlargement of the surface area, and loss of the migratory leading edge (Figure 2B). Particularly in PC3 cells, the actin fibers’ signal appears to be condensed in focal areas located in the basement of central areas of the cells. Similar results were obtained with the PC3 clones with knocked-down MK5 (Figure 2B, bottom panels), confirming that the effects of GLPG or MK5 depletion were similar in inducing reorganization of the actin filaments’ network.

### 3.3. The Motility Properties of MK5 Can Be Explained via Alterations of Focal Adhesion Complex

It has been recently reported that the TLK1 paralog, TLK2, could form a complex with Src and activate the EGFR/Src/FAK signaling pathway to promote the migration and invasion of breast adenocarcinoma and glioblastoma cells [19,20]. On the other hand, Yoshizuka et al. demonstrated that MK5 increases motility in endothelial cells by phosphorylating FAK on Y397 residue and enhances its localization to focal adhesions in advanced skin carcinogenesis [22]. Additionally, another group reported that recombinant MK5 can phosphorylate FAK, Src, and paxillin in IVK assays and form complexes with FAK and Src and that endogenous MK5 localizes to focal adhesions [26]. We hypothesized that TLK1 interaction with MK5 recruits it to focal adhesions where both TLK1 and MK5 may form a complex with either FAK or Src or Paxillin and transduce downstream signals. This process could form the basis for the visual effects shown in Figure 2. Furthermore, TLK1–MK5 colocalization with FAK may recruit it to the focal adhesions and activate FAK by Y397 auto-phosphorylation, and Tyr397 phosphorylation will recruit Src and activate it by Y416 autophosphorylation [28]. Activated Src can, again, phosphorylate FAK in Y861/Y925 and promote its full activation by creating binding sites for other adaptor proteins [29]. We thus set out to test the activation status (phosphorylation) of these key regulators/components of the focal adhesion complex. We first tested the pFAK-Y397 autophosphorylation (a marker of its intrinsic activity). However, treatment with GLPG of either LNCaP or PC3 showed no change in this phosphorylation (Figure 3A). Rather, an interesting result was observed for the Y861 phosphorylation, with LNCaP showing some loss and PC3 showing some gain. We should caution, however, that in LNCaP, there appeared to be also a slight loss of total FAK, and that the pY861 signal was sufficiently low to begin with in the control LNCaP, that the pY861 reduction in LNCaP cells may not be significant (despite its quantification on the right; see Figure 3B, left panel). In contrast, the increase in PC3 pY861 is highly significant, as its quantification shows (Figure 3B, right panel). The pY861 status was studied before in LNCaP and PC3 cells, and it was reported that FAK autophosphorylation (Y397) is adhesion dependent, whereas a second site of tyrosine phosphorylation, Y861, a Src-specific site, is uncoupled from adhesion-dependent signaling events [30]. Of course, the fact that pY861 does not correlate with adhesion events does not imply that it cannot promote these events, as its presence may be a necessary but insufficient modification for full FAK activation. In particular, FAK activation status can affect its interaction with Paxillin and its phosphorylation on numerous sites, including Y31 and Y118 [31,32]. The phosphorylation of paxillin on Y31 and Y118 leads to enhanced Rac activity and loss of RhoA activity, affecting adhesion and motility [32]. We thus probed for pPaxillin-Y118 in LNCaP and PC3 cells treated with GLPG. Again, the two cells appeared to show opposite effects by GLPG on Y118 phosphorylation: some signal loss in LNCaP and clear gain in PC3. Again, we should caution that the basal Y118 signal in LNCaP cell was very weak to begin with, and that the decrease in signal quantitated on the right (Figure 3C, left panel), despite its annotation as moderately significant, should be considered in the context of whether a “weak signal” becoming “negligible” can be taken as a possible indication of biological significance for motility. In contrast, the robust pY118 signal and its highly significant increase in PC3 cells can be clearly considered in light of its probable effect on adhesion and motility (or rather lack thereof), as manifested by the development of very visible large adhesion foci in PC3, either treated with GLPG or with loss of MK5 (Figure 2, Figure 3C (right panel) and Figure 4).

### 3.4. MK5 Could Promote Stress Fibers Formation through Its Effect on HSP27 Phosphorylation and Suppressing Its Actin Fibers’ “Capping Activity”

The TLK1–MK5 interaction may also promote cellular motility by remodeling actin fibers mediated through HSP27 phosphorylation, thus promoting actin polymerization. Cellular motility is generally orchestrated by the formation of protruding bodies (lamellipodia, filopodia, and invadopodia) which are rich in sub-membranous actin filaments. MK5 can regulate actin cytoskeletal reorganization through its most established substrate: HSP27. Non-phosphorylated HSP27 functions as a capping protein to inhibit F-actin polymerization, thus inhibiting stress-fiber formation [33,34,35,36,37,38]. Upon external stress-related stimuli, activated MK5 can phosphorylate HSP27 on S15, S78, and S82 residues, thus hindering its capping activity, and allows monomeric actin to polymerize into filamentous actin [23,24,25,39,40,41]. So, we tested whether the treatment of LNCaP and PC3 cells with GLPG affects HSP27 phosphorylation. In LNCaP cells, which clearly express vast amounts of HSP27 (much more than PC3), there was no clearly visible effect on pHSP27-S82. In contrast, there was a very significant loss of pHSP27-S82 in PC3 (Figure 3A,D). This may correlate well with the general loss of the actin fibers’ network that was observed in these cells after treatment with GLPG (Figure 2). Note that we were unable to detect any signals for pHSP27 S15 and S78 in either LNCaP or PC3 cells. 

Finally, we hypothesized that the activation of MK5 via its interaction with TLK1 via phosphorylation of S386 [7], which lies within the ERK3 binding module (reviewed in Reference [42]), may increase the association with ERK3 and promote its stabilization. ERK3 is an atypical MAP kinase, which is unstable in normal physiological conditions, as it is rapidly degraded by the ubiquitin proteasomal pathway [43,44]. Activated MK5 can reciprocally phosphorylate ERK3 and stabilize it [45]. Stabilized ERK3 may regulate MMP2, MMP9, and MMP10 expression by activating the steroid receptor coactivator SRC3 and, hence, is implicated in cell migration and invasion [46,47]. We thus tested the expression of ERK3, following the treatment with GLPG, typically determined by its stability. Indeed, treatment with GLPG caused a decrease in ERK3 expression in both LNCaP and PC3 at 3 µM and 5 µM concentrations, respectively (Figure 3A,E). Note that, as a control, we monitored whether GLPG affected TLK1/1B expression in these cells, and it did not (Figure 3A). Moreover, GLPG treatment did not affect the total MK5 level in these cell lines (Figure 3A). 

### 3.5. Relocalization of Paxillin to Focal Contacts of the Basal Surface Area of the Cells in PC3 Cells Treated with GLPG

The clear effect of GLPG on increasing phosphorylation of Paxillin Y118 suggested a very plausible explanation for the formation of very different and large focal adhesions centers at sporadic basal surface areas of PC3 cells (Figure 2). We thus set out to test whether Paxillin was indeed colocalized with these dense actin-rich domains. In PC3 cells treated with DMSO (vehicle control), staining with anti-Paxillin antibody resulted in rather diffused cytoplasmic staining presumably all along the basal contact surface area, with no distinct accumulation except from some puncta on the contour area of the cells, where it appeared to colocalize with the peripheral actin fibers’ network. After treatment with GLPG (particularly at the higher concentration), Paxillin staining appeared to remobilize from the peripheral areas (lamellipodia) to the basal contact side of the cells with the ECM, and more densely with the denser actin fibers foci (Figure 4).

### 3.6. The Loss of Invasive Properties of the Cells upon GLPG Treatment Correlate with Reduced MMPs Expression

As stated above, stabilized ERK3 via its feedback regulation by MK5 may regulate MMP2, MMP9, and MMP10 expression via activating the steroid receptor coactivator SRC3 and, hence, is implicated in cell migration and invasion [46,47]. The remarkable effects of GLPG in the aforementioned invasion assays, along with the reduced expression of ERK3, prompted us to check for the expression of MMPs, first at the transcript and then at the protein level in LNCaP and PC3 cells. With the main MMPs that we tested, GLPG caused a dose-dependent reduction of mRNA expression in both cell lines, which reached statistical significance for MMP9 in both LNCaP and PC3 and for MMP10 in PC3 cells (Figure 5A–C). Since the real test for expression, not considering their proteolytic activity, is at the protein level, we tested for MMPs by WB. Notably, MMP2 was not detectable in LNCaP (despite its qPCR signal) (Figure 5D), but it was clearly expressed in the more invasive PC3 cells, and its expression was progressively decreased after the treatment with GLPG (Figure 5D,E). MMP9 was decreased in a dose-dependent fashion in both lines (Figure 5D,F). In contrast MMP3/10, which was highly expressed particularly by the LNCaP cells, appeared unchanged in either cell line (Figure 5D), despite the modest decrease in mRNA expression (Figure 5C). Again, since pharmacologic inhibition can have off-target effects besides MK5, we tested the effect of knock down of MK5 in the PC3 shRNA clones. In all the clones, we verified a reduction in the expression of ERK3, attributable to its stabilizing effect of phosphorylation by MK5 (Figure 5G,H). MMP2 and MMP9 were likewise decreased in the MK5 knocked-down clones (Figure 5G,I,J). In contrast, the expression of MMP3/10 was unaffected (Figure 5G), similarly to the GLPG-treated PC3 cells.

### 3.7. Treatment with GLPG or J54 Reduces Experimental Metastases

Experimental metastasis was studied via an established model [48]. For this work, we primarily used the PC-3 cells because they generate reliable metastatic data, while other established PCa cell lines (e.g., LNCaP or DU145) are inconsistently or poorly metastatic in mice [49]. In the tail-vein injection model, the first site accessible to the tumor cells is typically the lungs. We have found evidence of metastases primarily in the lungs in virtually all the mice we have injected with PC3 cells. However, we found that, whereas in control mice there were typically 10–40 lung tumor nodules/mouse, (Figure 6A,B), mice treated within a day of inoculation with GLPG or J54 showed typically very few tumor nodules macroscopically. To be able to assess this at the microscopic level, the lungs were serially dissected in selected mice and examined histologically and by IHC. Some examples are shown in Figure 6C, where the H&E shows frequent tumors in untreated mice, mice subsequently treated with GLPG or J54 showed only sporadic areas with microscopic tumor colonies. Since administering the MK5 or J54 inhibitors was started one day after the tail-vein dissemination, it was not possible to establish if J54 or GLPG0259 affected the seeding, the migration and extravasation, or the motility/tissue-colonization of the injected cells. However, a staining for the proliferative marker Ki-67 (used to mark the colonies of PC3 cells) indicated that, most likely, the extravasation or seeding time from the circulating pool of PC3 cells was affected. We also noted that most tumor cells appeared to be proliferating (Ki-67+) in the GLPG-treated mice, as most of the cells in the tumor nodules appeared positive. The same may not be said for mice treated with J54, which showed few proliferating cells in the small patches of micro-metastases. This can readily be explained by the fact that J54, an inhibitor of TLK1, could affect additional cancerous pathways that are regulated by this kinase, such as the TLK1 > NEK1 > YAP nexus [17] or the AKTIP > AKT [18] pathway, thus resulting in overall suppression of tumors growth.

## 4. Discussion

For over 50 years since the Nobel Prize–winning research of Charles Huggins for his discovery that PCa cells require androgen for survival and proliferation, the majority of the research and treatment for PCa has evolved around androgen-deprivation therapy (ADT). In fact, for the practical management of PCa, it was subdivided into two phases of the disease: androgen sensitive (AS), which becomes indolent upon ADT, and androgen insensitive (AI or Castration-resistant prostate cancer (CRPC))). Once the cancer has become CRPC, there are no successful cures that add a decent quality and quantity of life, even considering the relative success of PARPi for the treatment of mCRPC patients with BRCAness features [50]. Like most other solid tumor types, PCa disease progression and death are primarily from the metastatic spread to bones and other organs, and elective sentinel lymph nodes dissection offers limited usefulness when combined with prostatectomy, as in many cases, the cancer has already spread to other sites at diagnosis. Treatments that can help curtail metastatic spread would be greatly beneficial, and the recent advent of Lutetium^177^–PSMA-617 radioligand for targeting CRPC metastases expressing PSMA holds the promise for improved outcome in a precision-medicine approach (VISION ClinicalTrials.gov (accessed on 17 November 2022) number, NCT03511664). 

Our work progressed from identifying the novel role of two protein kinases (TLK1 and MK5) and their potential downstream signaling mechanisms in promoting PCa cells motility and metastasis. Given that both kinases have specific pharmacologic inhibitors, we aimed to pursue if the inhibition of the TLK1–MK5 axis could bring any therapeutic benefit that will help to keep the tumor localized in preclinical studies. Specifically, we wanted to determine if J54 (TLK1 inhibitor) or GLPG (MK5 inhibitor), as well as a genetic approach to knock down TLK1 and MK5 with shRNA, could alter the motility, invasion, and ultimately metastatic properties of PCa cell lines. Having met with success for each of these goals, the next step was trying to understand the mechanisms underlying the pharmacologic effects resulting in reduced metastatic ability.

We first focused on motility and investigated the role of TLK1 > MK5 in the organization of actin fibers network at lamellipodia (the moving edge of the cells) by treating LNCaP and PC3 cells with GLPG. We found that, in fact, the phalloidin signal was relocalized from the dense staining at the periphery (lateral membrane contour) to much fewer and larger foci that appear to mark adhesion sites at the basal side of the cells in contact with the ECM surface, along with a change in morphology. This visual reorganization is suggestive of a change of actin fibers from motile structures at the leading edges to more solid contacts with substratum that could impair mobility. Notably, the pharmacologic effect of GLPG could be recapitulated by the depletion of MK5 in PC3 cells (Figure 2). We attempted to directly correlate the general loss of actin stress fibers with the known ability of MK5 to phosphorylate HSP27 (one of its best-known substrates). In LNCaP cells, there was massive expression of HSP27, and, as such, it would have been perhaps difficult to detect a change in S82 phosphorylation signal. However, this was observed in PC3 cells wherein GLPG caused a dose-dependent reduction of S82 phosphorylation (Figure 3). On the other hand, the relocalization of actin fibers to large basal layer focal adhesions suggested that, more likely, these are the reason for their reorganization and, consequently, reduced mobility. After testing the phosphorylation status of several components, and particularly kinases and their effectors, of the focal adhesion complex, we determined that the pFAK-Y861 and pPaxillin-Y118 were significantly increased in PC3 cells treated with GLPG. Activated FAK can recruit and phosphorylate paxillin on Tyr118 residue, which creates a binding sites for p130^Cas^-Crk and Dock180, which also play roles in cell migration [51]. Whether or not these changes are important for the alterations observed in Paxillin reorganization (Figure 4) will require much additional work: largely, confocal microscope and perhaps the use of phospho-specific antibodies instead of pan-paxillin and studying its colocalization with vinculin. Obviously, neither pPaxillin-Y118 nor FAK-Y861 can be a direct substrate of MK5 (a S/T kinase), but the increase in Y118 phosphorylation can certainly be explained via a priming phosphorylation event at any one of the many serine sites [52] on this complexly regulated focal adhesion protein for which just a few kinases have been established in its regulation, e.g., S273. For instance, S273 regulates Paxillin’s interaction with PAK1 at dynamic adhesion subdomains that correspond to regions at protrusive edges of the cell [53] that are also enriched in Src kinases. Whereas it was found that TLK2 promotes the activation of Src, EGFR, or FAK, there were no reports of direct interaction with these proteins at endogenous level [19], and we should mention that our attempts to demonstrate specific interactions with TLK1 by co-IP/MS identification have been mostly unsuccessful due to the sheer number of loosely associated proteins (unpublished). This approach was therefore replaced by our direct Interactome analysis [13], which revealed that the protein bound with the highest avidity in the study is Actin-Binding LIM protein 1 (ABLIM1). This is a LIM zinc-binding domain-containing protein that binds to actin filaments and mediates interactions between actin and cytoplasmic targets [54], suggesting that TLK1B can preferentially localize to focal adhesions and phosphorylate targets therein.

While the initial work was focused on trying to explain the motility changes obtained with GLPG and J54 treatments, the effects on the loss of invasive ability required additional studies. In our earlier study, we had identified S386 as one of three TLK1-mediated phosphorylation sites on MK5. As this site maps to the ERK3-interacting domain (reviewed in Reference [42]), and ERK3 is both a key activating kinase for MK5 but also its well-known reciprocal substrate, leading to its stabilization, we wondered if MK5 loss could affect ERK3 expression/stability. In fact, in PC3, loss of MK5 with shRNA caused a significant reduction of ERK3 signal (Figure 5G). In fact, ERK3 is an atypical MAP kinase, which is unstable in normal physiological conditions, as it is rapidly degraded by ubiquitin proteasomal pathway [43,44]. It was known that activated MK5 can phosphorylate ERK3 and stabilize it [45]. Stabilized ERK3 may then regulate MMP2, MMP9, and MMP10 expression by activating the steroid receptor coactivator SRC3 and, hence, is implicated in cell invasion [46,47] likely through proteolysis of the basement membranes. To test this hypothesis, we investigated the expression of these key metalloproteinases at both the mRNA and protein level in LNCaP and PC3 cells treated with GLPG. Indeed, some of these were moderately reduced in expression, particularly MMP2 and MMP9 (Figure 5A–F). Whether these are sufficient to explain the reduced capacity to migrate through Matrigel-coated inserts remains to be investigated more directly. 

Ultimately, we really wished to know if pharmacologic inhibition of the TLK1 > MK5 axis could really impact lung metastatic spread, and we resorted to the simplest test via the tail-vein injection method. Indeed, both GLPG and J54 resulted in a dramatic loss in lung tumor nodules, which, in some animals, were only visible in histologic section (if at all). This holds significant promise for future clinical translation of these results and confirms a recent report of the effectiveness of GLPG to suppress metastases in xenograft models of breast cancer and melanomas [21], although these authors adopted the previously held explanation that MK5/PRAK exerts its pro-motility effects through the regulation of mTORC1 [55]. In regard to the experimental metastases results, it was noticeable that GLPG did not reduce the size of the tumor nodules compared to untreated animals, but it clearly reduced the number of lung metastases. This may be used to argue against the mTORC1 pathway being involved, as this would likely also affect proliferation. In contrast, the TLK1 inhibitor J54 also reduced the size of the tumors—visibly so upon lung dissection—but also histologically for micrometastases that were generally negative for Ki-67 staining. This suggests that, in vivo, J54 may modulate also the proliferation of cancer cells, a property that we attributed earlier to the TLK1 > NEK1 > YAP1 axis [17], largely affects the contact-inhibition features (mechanotransduction) of the cells [56]. In conclusion, we have demonstrated the critical importance of controlling the TLK1 > MK5 axis for reducing PCa metastatic spread, perhaps offering a novel point of attack to combat this incurable stage of the disease. A schematic model of the mechanisms of TLK1–MK5 axis in pro-metastatic involvement is provided in Figure 7.

## 5. Conclusions

There is currently no cure for PCa that has progressed to the mCRPC stage, with metastases particularly to the bones being the ultimate cause of morbidity and mortality. Few therapies to suppress metastatic disease are known for mCRPC, but GLPG, a rather specific MK5 inhibitor, has recently entered the arena as a metastasis suppressor particularly in a xenograft model of breast cancer. Our work has now shown this for two PCa cell lines, and, in particular, it has shown a strong reduction of lung tumors for the highly invasive and metastatic cell line PC3. We further studied, both at the molecular and cellular level, whether this is largely due to discrete changes in cytoskeletal organization and focal adhesions, as well as reduced ECM invasiveness through the reduction of MMPs’ expression. All of this is hopeful for the next translation of GLPG or J54 into possible clinical trials.

## Figures and Tables

**Figure 1 cancers-14-05728-f001:**
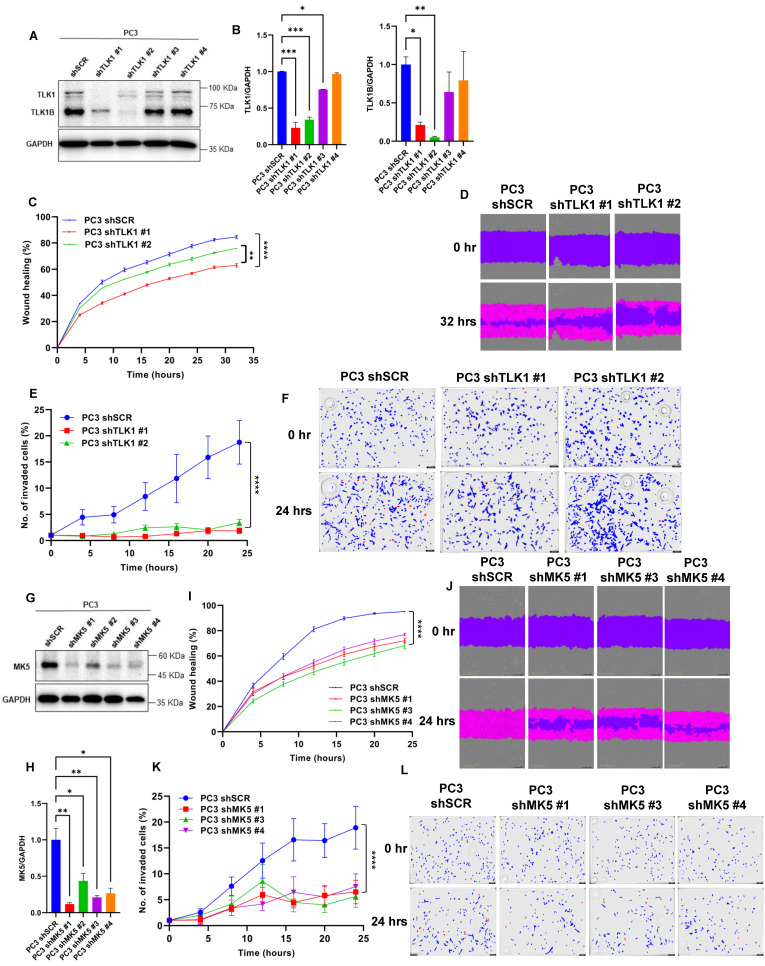
**Genetic depletion of TLK1 or MK5 leads to reduced motility and invasion of PC3 cells.** (**A**) Stable knocked down of TLK1 and TLK1B upon either scrambled (shSCR) or four different TLK1-specific shRNA treatments in PC3 cells transduced with lentivirus. Image is the representation of two independent experiments (*n* = 2). (**B**) Quantification of TLK1 (left panel) and TLK1B (right panel) level upon TLK1 knockdown normalized to GAPDH in PC3 cells. (**C**,**I**) Scratch wound repair assay was conducted to determine the migration rate among (**C**) PC3 shSCR, PC3 shTLK1 #1, and PC3 shTLK1 #2; and (**I**) PC3 shSCR, PC3 shMK5 #1, PC3 shMK5 #3, and PC3 shMK5 #4 cells by plotting relative wound density against different time points; *n* = 6–12 biological replicates were used for each cell line. (**D**,**J**) Image representation of the scratch wound repair assay. (**E**,**K**) Trans-well invasion assay was conducted to determine cellular invasion capabilities among (**E**) PC3 shSCR, PC3 shTLK1 #1, and PC3 shTLK1 #2; and (**K**) PC3 shSCR, PC3 shMK5 #1, PC3 shMK5 #3, and PC3 shMK5 #4 cells at different time points, using Matrigel coating. The invasion rate was determined by plotting the total object phase area against time; *n* = 5–7 biological replicates were used for each cell line. (**F**,**L**) Image representation of the trans-well invasion assay. Blue color represents the cells in the top chamber, and red color represents the cells that invaded into the bottom chamber. (**G**) Stable knocked down of MK5 upon either scrambled (shSCR) or four different MK5-specific shRNA treatments in PC3 cells transduced with lentivirus. Image is the representation of two independent experiments (*n* = 2). (**H**) Quantification of MK5 level upon MK5 knockdown normalized to GAPDH in PC3 cells. One-way ANOVA, followed by Tukey’s post hoc analysis, was used for multiple group comparison. Groups that are significantly different from each other are indicated by asterisks: * = *p* < 0.05, ** = *p* < 0.005, *** = *p* < 0.0005, and **** = *p* < 0.0001. Error bar represents standard error of the mean (SEM). The whole blots of Figure 1 could be found in Appendix A.

**Figure 2 cancers-14-05728-f002:**
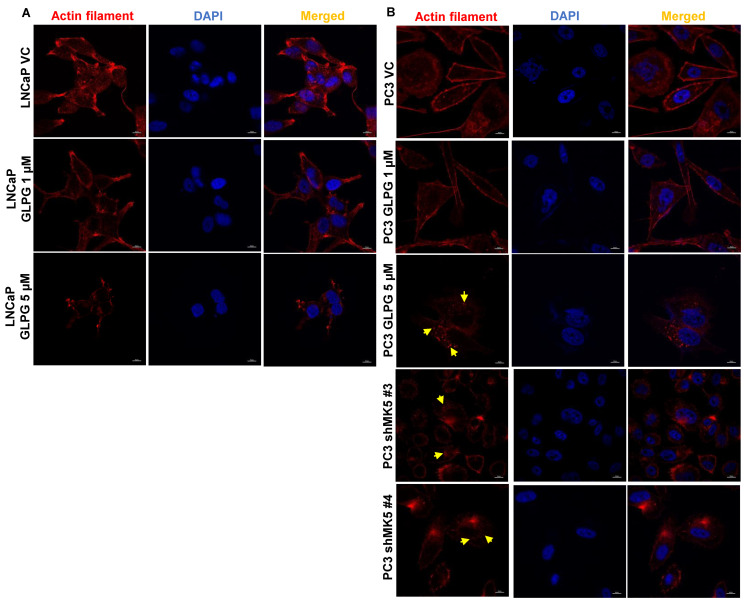
**MK5 inhibition/depletion leads to the reduction and remodeling of actin filamentation in LNCaP and PC3 cells.** Rhodamine-conjugated phalloidin staining to determine the actin filamentation of (**A**) LNCaP cells treated with different doses of GLPG, and (**B**) PC3 cells treated with different doses of GLPG or stable genetic depletion of MK5. VC = vehicle control (DMSO); GLPG = GLPG0259 (MK5 inhibitor). Scale bar = 10 µm. The arrow represents the condensed actin foci in the basal surface of the cells. Images are the representation of three independent experiments (*n* = 3).

**Figure 3 cancers-14-05728-f003:**
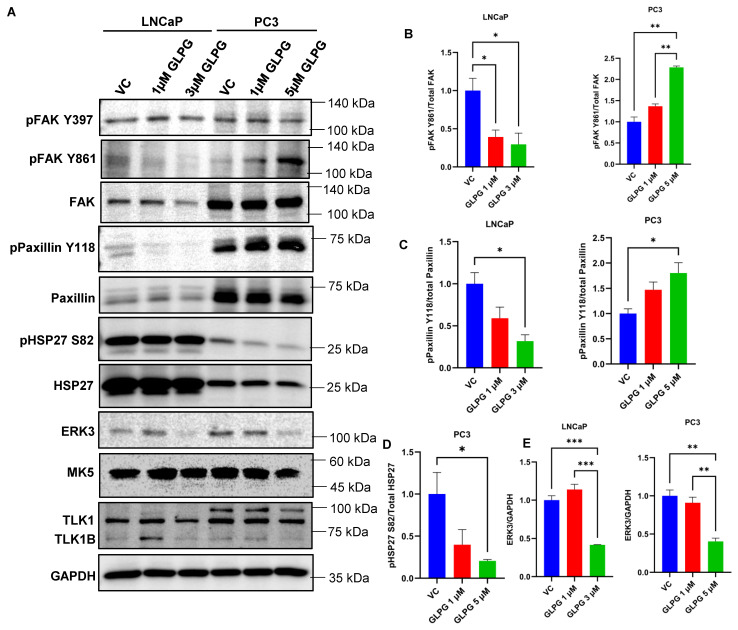
**MK5 regulates cellular motility through post-translational modification of focal adhesion proteins and HSP27, as well as regulation of ERK3 stability in LNCaP and PC3 cells.** (**A**) Immunoblots of the indicated proteins in LNCaP and PC3 cells treated with different concentrations of GLPG0259. GAPDH was used as a loading control. Images are the representation of three independent experiments (*n* = 3). (**B**–**E**) Quantifications of (**B**) Phospho-FAK Y861 level normalized to total FAK; (**C**) Phospho-Paxillin Y118 level normalized to total paxillin; (**D**) Phospho-HSP27 S82 normalized to total HSP27; and (**E**) ERK3 level normalized to GAPDH in LNCaP and PC3 cells. VC = vehicle control (DMSO). One-way ANOVA, followed by Tukey’s post hoc analysis, was used for multiple group comparison. Groups which are significantly different from each other are marked by asterisks: * = *p* < 0.05, ** = *p* < 0.005, and *** = *p* < 0.0005. Error bar represents standard error of the mean (SEM). The whole blots of Figure 3 could be found in Appendix A.

**Figure 4 cancers-14-05728-f004:**
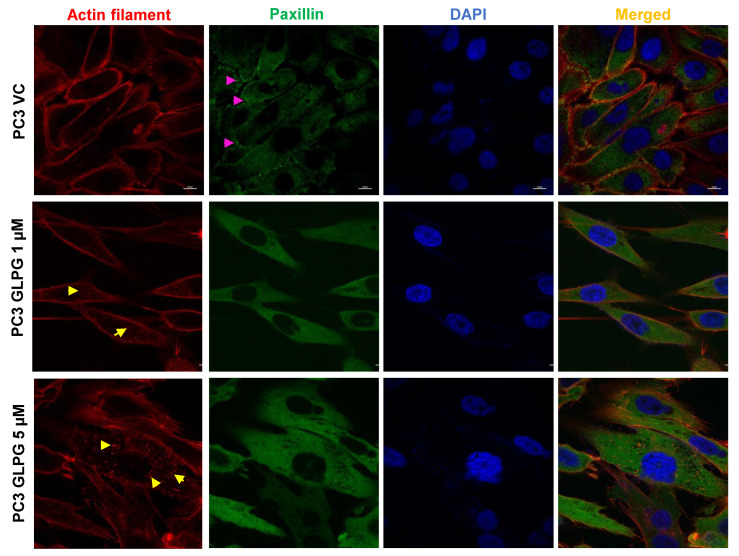
**MK5 regulates the formation of focal adhesions in the lamellipodial extensions of migratory PC3 cells.** Immunostaining of paxillin along with rhodamine phalloidin–mediated actin staining to determine the localization of focal adhesions in PC3 cells treated with different doses of GLPG. Images are the representation of three independent experiments (*n* = 3). Scale bar = 10 µm. The pink arrow represents paxillin localization in the lamellipodial extensions in the vehicle control (VC) treated cells, while the yellow arrow represents the condensed actin foci in the basal surface of the GLPG-treated cells.

**Figure 5 cancers-14-05728-f005:**
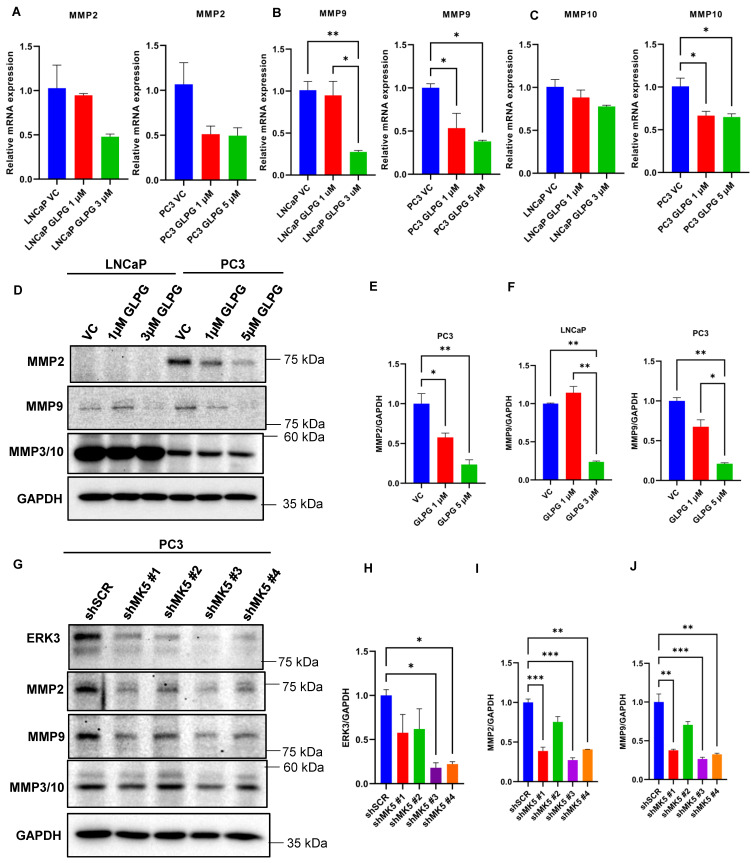
**MK5 regulates matrix metalloproteinases’ expression in PC3 cells.** (**A**–**C**) mRNA expression of several matrix metalloproteinases (MMPs): (**A**) MMP2, (**B**) MMP9, and (**C**) MMP10 in LNCaP and PC3 cells treated with different concentrations of GLPG. GAPDH mRNA was used as an internal control. (**D**) Immunoblots of the indicated proteins in LNCaP and PC3 cells treated with different concentrations of GLPG. GAPDH was used as a loading control. Images are the representation of three independent experiments (*n* = 3). (**E**,**F**) Quantifications of (**E**) MMP2, and (**F**) MMP9 level normalized to GAPDH in LNCaP and PC3 cells. (**G**) Immunoblots of the indicated proteins in PC3 cells upon MK5 depletion. GAPDH was used as a loading control. (**H**–**J**) Quantifications of (**H**) ERK3, (**I**) MMP2, and (**J**) MMP9 level normalized to GAPDH in PC3 cells upon MK5 depletion. VC = vehicle control (DMSO); shSCR = scrambled shRNA. One-way ANOVA, followed by Tukey’s post hoc analysis, was used for multiple group comparison: * = *p* < 0.05, ** = *p* < 0.005, and *** = *p* < 0.0005. Error bar represents standard error of the mean (SEM). The whole blots of Figure 5 could be found in Appendix A.

**Figure 6 cancers-14-05728-f006:**
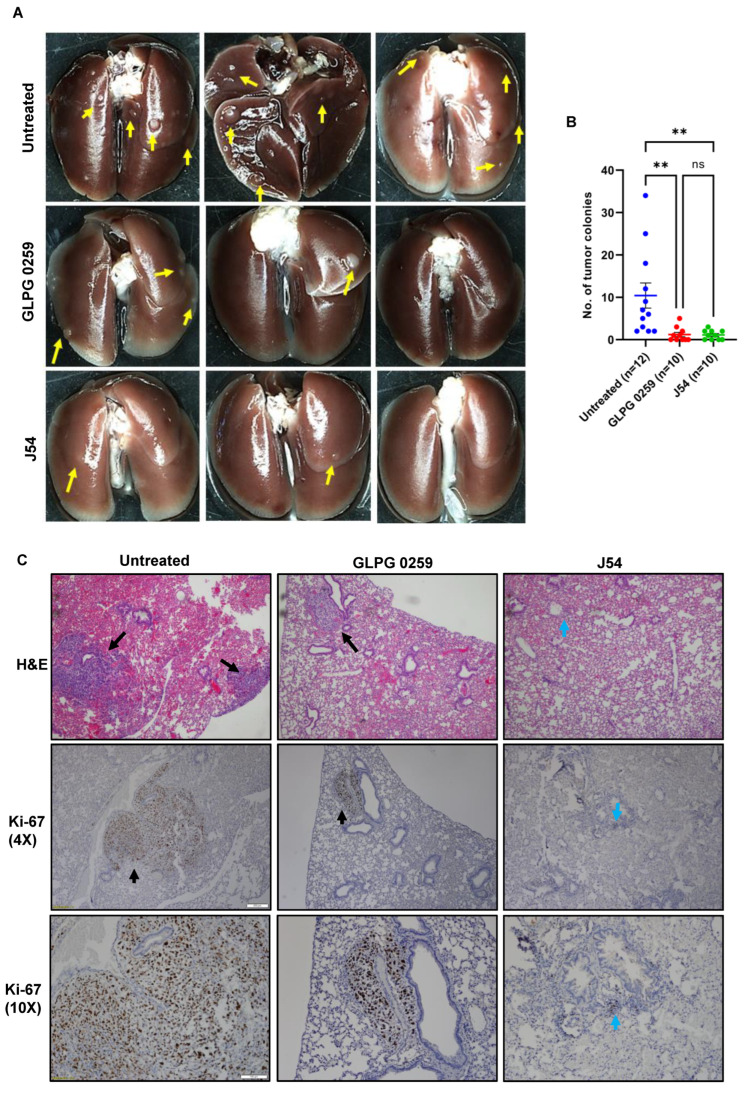
**TLK1 or MK5 inhibition impairs lung metastasis of PC3 cells in vivo.** Five-week-old SCID beige mice were injected intravenously via the tail vein with 1 × 10^6^ parental PC3 cells and treated with either GLPG0259 or J54 intraperitoneally twice a week for 7 weeks. Lung metastases were compared among the untreated, GLPG-treated, or J54-treated mice. (**A**) Four representative images from each group are shown. (**B**) Data are presented as the number of metastatic colonies in the lung. (**C**) H&E and Ki-67 immunostaining shows lung nodules and proliferative tumor cells, respectively. Scale bar: 4× = 200 µm; 10× = 100 µm. One-way ANOVA, followed by Tukey’s post hoc analysis, was used for multiple group comparison: ** = *p* < 0.005; ns = not significant. Error bar represents standard error of the mean (SEM). The yellow and black arrows represent macroscopic nodules, whereas the blue arrows represent the presents of microscopic metastatic lesions.

**Figure 7 cancers-14-05728-f007:**
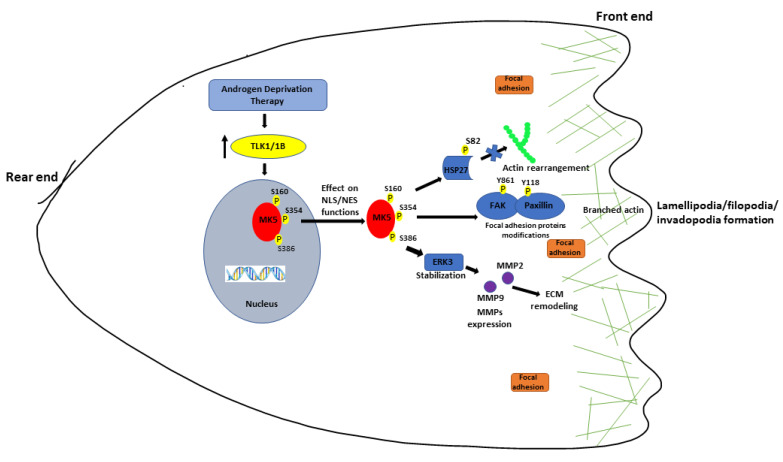
**A schematic diagram showing the mechanisms of TLK1 > MK5 signaling in promoting prostate cancer migration and invasion.** Combined with our previous findings, TLK1-mediated phosphorylation of MK5 in three novel residues (S160, S354, and S386) may promote the shuttling of MK5 out of the nucleus, where it exerts its pro-metastatic function by modulating the phosphorylation of focal adhesion proteins (pFAK Y861 and pPaxillin Y118) and HSP27 (pHSP27 S82) that enhances actin filamentation and reorganizations in the leading edge of the cells. Furthermore, TLK1 > MK5–mediated ERK3 stabilization stimulates several MMPs’ (MMP2 and MMP9) expression, which enhances invasive capacities of the PCa cells. Co-ordinated effects of these signaling events finally orchestrate the functional metastasis of PCa cells.

**Table 1 cancers-14-05728-t001:** Primer sequences.

Primer	Sequence-5′-3′
MMP2-F	5′-TCCACCACCTACAACTTTGAG-3′
MMP2-R	5′-GTGCAGCTGTCATAGGATGT-3′
MMP9-F	5′-ACATCGTCATCCAGTTTGGTG-3′
MMP9-R	5′-CGTCGAAATGGGCGTCT-3′
MMP10-F	5′-GGAGACTTTTACTCTTTTGATGGC-3′
MMP10-R	5′-AGCAACGAGGAATAAATTGGTG-3′
GAPDH-F	5′-ACATCGCTCAGACACCATG-3′
GAPDH-R	5′-TGTAGTTGAGGTCAATGAAGGG-3′

## Data Availability

All cell lines, reagents generated in this study, and full data files are available upon request.

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
