# Peer review of "The TLK1–MK5 Axis Regulates Motility, Invasion, and Metastasis of Prostate Cancer Cells"

_cancers, 2022, doi:10.3390/cancers14235728_

Round 1
Reviewer 1 Report
In this study, the authors investigated the effect of the TLK1-MK5 axis on the motility, invasion, and metastasis of prostate cancer cells. Using inhibitors or shRNAs of TLK1 and MK5 reduced the migration and invasion of PC3 cells in vitro and lung metastasis of PC3 cells in vivo. The authors also elucidated the underlying mechanisms that the TLK1-MK5 axis may regulate the reorganization of the actin fibers at lamellipodia and the focal adhesions network, in conjunction with increased expression of some MMPs that can affect penetration through the ECM.
Below are some questions that the authors need to address.
- Please upload high-resolution figures. The red staining in Fig. 1F&L is invisible.
- Again, please replace the current Fig. 2 with high-resolution images.
- Since Fig. 2 showed similar phenotype changes in PC3 and LNCaP cells, did authors perform in vitro assays in LNCaP cells to confirm the results in PC3 cells in Fig.1?
- A co-IP experiment is necessary to support that TLK1 and MK5 form a complex with either FAK or Src or Paxillin.
- In the manuscript, the authors sometimes use “GLPG”, and sometimes use “GLPG0259”. Please be consistent.
- In this study, the biological effects of the TLK1-MK5 axis on motility were mainly tested in the PC3 cell line. The authors need to verify the conclusion in at least one more cell model.
Author Response
1.We have enlarged a little panels F and L. It is not a matter of resolution but rather magnification. We believe that now the bottom plane cells (red) should be visible even with the multi-panel figure.
- 2. All figures were converted to TIF images.
- 3. As we have explained in the text, LNCaP cells do not tolerate loss of TLK1 via shRNA; we have not been able to obtain a stable line, nor a transient knockout. On the other hand, we have already published that GLPG strongly suppresses motility of LNCaP cell in the MOLONC paper.
- 4. This experiment cannot be done with any confidence: Whereas it was reported that TLK2 promotes the activation of SRC, EGFR or FAK, there were no reports of direct interaction with these proteins (https://www.ncbi.nlm.nih.gov/pmc/articles/PMC5064015/), and we should report that our attempts to demonstrate specific interactions with TLK1 by coIP/MS identification have been mostly unsuccessful due to the sheer number of loosely associated proteins (unpublished). This approach was therefore replaced by our direct Interactome analysis (https://www.ncbi.nlm.nih.gov/pmc/articles/PMC5462085/), which revealed that the protein bound with the highest avidity in the entire report is:
Actin Binding LIM protein 1 (ABLIM1). This is a LIM zinc-binding domain-containing protein that binds to actin filaments and mediates interactions between actin and cytoplasmic targets.
Therefore, almost anything associated with actin filaments and focal adhesions, including paxillin, FAK, and SRC, can potentially be found in a large complex with TLK1, depending on the washing conditions of the coIP. We have added text in the Discussion to emphasize this fact.
It is important to point out that our identification of the interaction between TLK1 and MK5 via the Interactome study, could be initially verified by direct association of the two recombinant proteins even before coIP analysis from cell extracts under stringent conditions.
On the other hand, the interaction of MK5 with SRC and FAK was previously reported by others.
- 5. We have made the replacement in the manuscript.
- 6. As explained above, shRNA to TLK1 is not tolerated in LNCaP cells, and several other cancer cell lines we have tested. However we have already published the effect of GLPG treatment in PC3, DU145, LNCaP, and C42B PCa cells.
Reviewer 2 Report
Well analysis at the molecular and cellular level of metastatic prostate cancer. In clinical, some hormone naïve patients under androgen deprivation therapy still progress rapidly. In this study, the tousled-like kinase 1 (TLK1) and MAPK-ac- 22 tivated protein kinase 5 (MK5) pathway may gave physician more information about metastatic prostate cancer.
Author Response
We thank the reviewer for the generous evaluation of the manuscript. We note that no suggestions for improvement were included.
Round 2
Reviewer 1 Report
My questions are addressed well.